# Impact of Titanium Addition on Microstructure, Corrosion Resistance, and Hardness of As-Cast Al+6%Li Alloy

**DOI:** 10.3390/ma16072671

**Published:** 2023-03-27

**Authors:** Marcin Adamiak, Augustine Nana Sekyi Appiah, Anna Woźniak, Paweł M. Nuckowski, Shuhratjon Abdugulomovich Nazarov, Izatullo Navruzovich Ganiev

**Affiliations:** 1Materials Research Laboratory, Faculty of Mechanical Engineering, Silesian University of Technology, 18A Konarskiego Street, 44-100 Gliwice, Poland; 2Department of Machinery and Devices, Engineering and Technological Faculty, Technological University of Tajikistan, Dushanbe 734061, Tajikistan; 3Nikitin Institute of Chemistry, National Academy of Sciences of Tajikistan, Dushanbe 734063, Tajikistan

**Keywords:** grain boundaries, microsegregation, SEM observations, EDS analyses, microhardness

## Abstract

Aluminum–lithium alloys have the potential for use in aerospace applications, and improving their physical, mechanical, and operational characteristics through alloying is a pressing task. Lithium, with a density of 0.54 g/cm^3^, enhances the elastic modulus of aluminum while reducing the weight of the resulting alloys, making them increasingly attractive. Adding transition metal additives to aluminum alloys enhances their strength, heat resistance, and corrosion resistance, due to their modifying effect and grain refinement. The study aimed to investigate the impact of titanium content on the microstructure, corrosion resistance, and hardness of Al-Li alloys. Four alloys were prepared with varying amounts of titanium at 0.05 wt%, 0.1 wt%, 0.5 wt%, and 1.0 wt%. The results showed that the microstructure of the alloy was modified after adding Ti, resulting in a decrease in average grain size to about 60% with the best refinement at 0.05 wt% Ti content. SEM and EDS analysis revealed an irregular net-shaped interdendritic microstructure with an observed microsegregation of Al_3_Li compounds and other trace elements at the grain boundaries. The samples showed casting defects due to the high content of Li in the alloy, which absorbed air during casting, resulting in casting defects such as shrinkage holes. The corrosion resistance test results were low for the samples with casting defects, with the least resistance recorded for a sample containing 0.1 wt% Ti content, with more casting defects. The addition of Ti increased the microhardness of the alloy to an average of 91.8 ± 2.8 HV.

## 1. Introduction

In the last decade, the development of new materials such as composites has increased significantly, but metals and alloys continue to be the primary building materials used in manufacturing machinery, equipment, construction, transportation, and communication. Therefore, improving upon the performance of metals and alloys is crucial not only to minimize economic losses but also to offer more technical solutions [1,2,3].

Aluminum–lithium alloys are a new type of aluminum system that possess an impressive combination of mechanical properties such as low density, elevated modulus of elasticity, and adequate strength. The increased interest in these alloys stems from the fact that lithium has a low density of approximately 0.54 g/cm^3^, resulting in a 3% reduction in aluminum density and a 5% increase in Young’s modulus for each percentage of lithium added. Despite these advantages, Al-Li alloys exhibit a marked anisotropy in mechanical properties and poor ductility. One possible explanation for their brittle behavior is their high sensitivity to harmful impurities [4].

In the pursuit of enhancing weight efficiency in the aviation and space sector, the use of materials with lower density is a key focus. Adoption of such materials can lead to reduced fuel consumption, extended flight ranges, increased passenger capacity, and a larger payload volume [5,6]. According to research, the best approach to reducing weight is by decreasing density, and then increasing strength and stiffness. The inclusion of 1% lithium, which is the lightest metal, results in a decrease of around 3% in density and a corresponding increase in elastic modulus of about 8% [7]. The first aluminum alloy that contained lithium was referred to as the Scleron alloy, which was developed in Germany [8], The composition of the alloy consisted of Al-12Zn-3Cu-0.6Mn-0.1Li, however, it was no longer produced due to its lack of superior properties compared to conventional aluminum alloys.

The development of aluminum–lithium alloys began in the 1950s by Alcoa in the United States. They created the high-strength alloy 2020 in 1957, which was composed of Al-4.5Cu-1.1Li-0.5Mn-0.2Cd on average [9]. To address the concerns with this alloy, the Al-Cu-Mg-Li-Cd system aluminum–lithium alloy VAD 23 was developed in 1960 [10]. The alloy in question had a reduced density of 3–5% and an increased elastic modulus of 3–5% compared to alloy 2020. In 1965, a team of researchers under the leadership of I. N. Friedlander discovered that the heat treatment of alloys in the Al-Mg-Li system resulted in a significant strengthening effect in a wide concentration range. They found that the addition of lithium, which has a lower modulus of elasticity than aluminum, increased the modulus by 3.8% in the Al-Mg-Li system alloys, a phenomenon known as the “Friedlander effect” [11]. Despite this, the corrosion resistance of the ternary alloys in the Al-Li system was low. The addition of manganese and zirconium improved it, leading to the development of alloy 1420 in 1969, which is the lightest aluminum alloy [12]. Additionally, it can be hardened through air cooling and artificial aging [13]. Despite operating in harsh environments and at various latitudes, aircraft built with alloy 1420 performed reliably without any reported corrosion damage [14]. The A.I. Mikoyan Development Design Office successfully made the MiG-29M aircraft with welded sealed tanks and a cabin using alloy 1420 in the 1980s. The aircraft’s weight was reduced by 24%, mostly attributed to the alloy’s lower density [15].

New alloys were created based on alloy 1420 including 1421, 1423, and alloys incorporating scandium [16,17,18,19]. To enhance the production feasibility, flexibility properties, and resistance to breaking of first-generation alloys [4,20,21], additional research was conducted on aluminum–lithium alloys in three distinct areas [22]: (1) Modifying the chemical composition of the alloys incorporating alloying elements such as zirconium and scandium to impact the grain structure, and silver and zinc to affect the strengthening phases during heat treatment; (2) Enhancing the manufacturing processes at every stage of producing semi-finished products, including homogenization methods, hot deformation and intermediate heat treatment during cold strain; (3) Developing multi-stage heat treatment techniques that specifically target the main resource characteristics (fatigue life and crack resistance) of the complex alloys. The second generation of aluminum–lithium alloys was developed in the United States and Europe in the 1970s to 1980s and in the Soviet Union in the 1980s to 1990s with the objective of obtaining alloys that were 8–10% lighter (and stronger) than conventional alloys of similar nature by adding lithium, copper, and small amounts of manganese and zirconium to control the grain size [23,24,25]. The most highly regarded second-generation aluminum–lithium alloys globally are 2090 and 8090 [26,27]. Following the success of alloy 1420, the second generation of aluminum–lithium alloys was developed, including alloys 1430 and 1441 in the Al-Cu-Mg-Li system, alloys 1450, 1451, and 1460 in the Al-Cu-Li system, and alloy 1424 in the Al-Mg-Li system. The primary goal of these alloys was to offer a range of operational qualities, including improved fracture toughness, crack resistance, and corrosion resistance matching the level of alloy 1420 [28,29,30,31,32,33].

Generation III aluminum–lithium alloys, which are in the process of being developed, include elements such as silver, zinc, indium, cerium, and tin to improve properties including corrosion resistance, fracture toughness, and weldability [34].

The addition of titanium as an alloying element has been found to improve the corrosion resistance, tensile strength, and fatigue strength of the materials. Optimal titanium contents of around 0.5 wt% have been reported in the literature for these improvements [35,36]. However, further research is needed to fully understand the effect of titanium on the properties of Al-Li alloys and to identify the optimal titanium content for different applications. This research studies the effects of different concentrations of titanium in an Al-Li alloy, on the microstructural evolution, corrosion resistance, and hardness.

## 2. Materials and Methods

The alloys were created using aluminum grade A995 (GOST 110669–74), lithium-LE1 (GOST 8774-75), and titanium-VT (GOST 19807-91). The alloys were synthesized in corundum crucibles using a resistance furnace heated to 750–850 °C, with an aluminum–titanium master alloy and a flux layer of NaCl-32.5 and KCl-32.5. Cylindrical samples, measuring 8–10 mm in diameter and 60–100 mm in length, were cast from the melted material for examination of microstructure and corrosion–electrochemical properties. Alloys with lithium were prepared using a vacuum oven-resistance type SNVE 1.3.1/16 in a helium atmosphere at a pressure of 0.5 MPa. Alloy blending took into account metal waste, and the composition was selectively controlled by chemical analysis and weighing of samples before and after fusion. The difference in weight before and after melting was less than 2% (relative). The investigated samples consisting of Al+6%Li with the addition of varying amounts of Ti as used to create the studied samples are listed in Table 1.

The cross-section of the investigated samples for metallographic tests was prepared using an automated grinding and polishing system. The specimens underwent a thorough metallographic preparation, including grinding with SiC papers, polishing with a coarse diamond suspension, and achieving a mirror finish with 0.04 μm colloidal silica. Finally, the polished surfaces were etched with Keller’s reagent, a mixture of nitric acid (2.5 vol.%), hydrochloric acid (1.5 vol.%), hydrofluoric acid (1.0 vol.%), and distilled water.

The microstructural evaluation was carried out using a combination of light microscopy, AxioVision (ZEISS, Jena, Germany), and scanning electron microscopy (SEM), with SEM being performed using a Zeiss Evo MA 15 series instrument equipped with an X-ray energy dispersive spectrometry (EDS) system.

Phase analysis was completed using X-ray diffraction with a PANalytical X’Pert Pro diffraction system (Panalytical B.V. (currently: Malvern Panalytical Ltd.), Almelo, The Netherlands) that was equipped with a cobalt anode lamp (KαCo λ = 0.179 nm), powered by voltage 40 kV, with the filament current intensity = 30 mA. The X-ray diffraction measurements were performed in the Bragg–Brentano geometry in the angular scope 30–110° 2θ with the step 0.05° and the step count time 100 s. The obtained diffractograms were analyzed by means of the X’Pert High Score Plus software (v. 3.0e) with a dedicated Inorganic Crystal Structure Database—ICSD (FIZ, Karlsruhe, Germany).

The Vickers hardness of the surface layers was measured using FM-ARS 9000 (Future Tech Corporation, Tokyo, Japan) hardness tester with a load of 4.9 N. The microhardness measurements were conducted on the XY cross-section of the samples following metallographic polishing. A total of 64 measurements were taken over a 1.4 mm × 1.4 mm area for each sample, in the form of an 8 × 8 matrix with an evenly spaced horizontal and vertical displacement of 0.18 mm. This was completed to determine the hardness of the surface, accounting for all surface conditions. Subsequently, 10 unevenly spaced points were selected and measured for each sample, avoiding surface imperfections that could interfere with the indenter. The microhardness measurement results from these methods would later provide insight into how surface imperfections affect the samples’ hardness.

The corrosion resistance of the test samples was evaluated using potentiodynamic testing, where anodic polarization curves were recorded. The measurements were carried out using a test setup that included an Atlas 0531 EU potentiostat (ATLAS-SOLLICH, Rębiechowo, Poland), a computer with AtlasLab software to save and analyze the recorded curves, and a three-electrode system that consisted of a platinum wire (auxiliary electrode), a silver/silver chloride electrode (Ag/AgCl) (reference electrode), and the test samples (anode electrode). The first step was to measure the open circuit potential (E_ocp_) for a period of one hour. Once the E_ocp_ was established, the corrosion test was initiated using the initial potential, which was calculated using Equation (1).
E_init_ = E_ocp_ − 100 mV(1)

The samples were then polarized by either reaching a potential of 2 V or by measuring the current density at 1 mA/cm^2^ along a range of −1 to +1 V at a scanning rate of 1 mV/s. The corrosion potential, Ecorr, was calculated from the results of the polarization measurements. The resistance to corrosion, Rp, was calculated using Tafel’s method. The electrochemical experiments were conducted in a solution of 3% NaCl at a temperature of 24 ± 1 °C.

## 3. Results and Discussion

### 3.1. Microstructure

The microstructure of the unmodified Al-Li alloy is displayed in Figure 1. It is evident that the typical casting process yields eutectic structures and imperfections, such as shrinkage holes. This occurrence is a result of the high concentration of Li in the alloy, causing air absorption during the casting process [37]. The formation of gas holes and microcracks was observed in Figure 1a. The structure of the as-cast alloy was comprised of a coarse dendritic network and a considerable amount of non-equilibrium eutectic phases found in the grain boundaries and the interdendritic region, depicted in Figure 1b.

The addition of titanium to the alloy caused a modification in grain size as the titanium content was increased. The polarized micrographs of the grains of the samples with different titanium contents are shown in Figure 2. The grain size distribution was analyzed, and the results are presented in statistical distribution plots and box and whisker charts in Figure 3. Figure 3a,b shows the grain size distribution and statistical analysis of the base Al-Li alloy without titanium. The average grain size was 355.2 µm, with a minimum of 36.6 µm and a maximum of 962.8 µm. The addition of 0.05 wt% titanium to the alloy in sample T1 resulted in smaller average grain sizes than the base sample. Figure 3c,d shows the grain size distribution of sample T1, with an average grain size of 222.4 µm, a minimum of 87.4 µm, and a maximum of 534.9 µm. However, further increasing the titanium content from 0.05 wt% to 0.1 wt% in sample T2 resulted in larger grain formation in the material. As shown in Figure 3e,f, the average grain size for sample T2 was 364.4 µm, with a minimum of 97.7 µm and a maximum of 1061.4 µm. The grain sizes continued to increase as the titanium content was further increased to 0.5 wt% in sample T3, as seen in Figure 3g,h with an average grain size of 454.3 µm, a minimum of 104.9 µm, and a maximum of 1456.9 µm. Similarly, after increasing the titanium content to 1.0 wt% in sample T4, the average grain size also increased. The mean grain size for this sample was 458 µm, with a minimum of 66.5 µm and a maximum of 1791.8 µm.

After alloying with titanium, the alloy is observed to have a refined crystal structure. Titanium has a dense hexagonal crystal structure with an atomic radius of 1.468 × 10^−10^ m. This enables it to form stable compounds with carbon and oxygen and the precipitation of nucleants such as Al_3_Ti. The Ti element is known for being an effective inoculant for refining the grain of aluminum alloys. However, some studies [38,39,40] have contended from the peritectic perspective that the addition of Ti can lead to satisfactory grain refinement unless it is added up to the maximum solubility limit in the α-Al matrix, triggering a peritectic reaction. This gives a possible explanation for the observed reduction in grain sizes with lower contents of Ti element in samples T1 and T2, and the gradual increase in the grain sizes as the Ti content was increased for samples T3 and T4. As reported by the research work [41], the peritectic reaction fosters the nucleation of the intermetallic compound Al_3_Ti, which enhances the mechanical performance of the alloy.

Figure 4 presents the chemical composition of the alloy after incorporating the titanium alloying element (Sample T1) as determined by energy-dispersive X-ray spectrometry (EDS). The analysis of the surface shows it is mainly composed of Al and Li, with small amounts of other elements from the casting process such as Mg, Fe, and Zr. A closer examination of the grain boundary, shown in Figure 5, indicates that there is a segregation of chemicals at the interdendritic phase. With regards to the EDS data and atom %, the constituents of the examined segregation were mostly made up of Li, with a relatively lower content of Al along with small amounts of Mg, Fe, or Zr. Figure 6, taken using scanning electron microscopy (SEM), reveals that the interdendritic phases have an irregular, net-shaped appearance. A large amount of Li caused the samples to absorb air during casting, resulting in the formation of defects such as shrinkage holes. These defects were more noticeable in sample T2 compared to sample T1. As the Ti content increased in samples T3 and T4, the shrinkage holes became bigger and deeper. The existence of casting defects such as microcracks and porosity can decrease the material’s resistance to corrosion [42]. The EDS maps in Figure 7 show the distribution of alloying elements, inhomogenously distributed within the material. Li was too light to be detected by the EDS during the maps generation in Figure 7.

The presence of microsegregation at the grain boundaries is attributed to non-equilibrium segregation of solute elements at these locations. This phenomenon occurs when enough vacancy–solute complexes form. However, rapid cooling of the melt through a large temperature range can cause the equilibrium concentration of these complexes to decrease and prevent establishment of true equilibrium concentration, except at sites (sinks) where vacancies are absorbed [43]. The sinks in the material include surfaces and interfaces between grain boundaries. The movement of vacancies towards these sinks is facilitated by the formation of a vacancy concentration gradient in the rapidly cooled melt. This concentration gradient enables solute atoms to be carried and deposited at the sinks. The accumulation of excess solute atoms at the grain boundary leads to non-equilibrium segregation at the grain boundaries [44,45,46].

### 3.2. X-ray Diffraction (XRD)

The X-ray diffraction (XRD) analysis was conducted to determine the phases present in the alloy after titanium modification. The XRD patterns from the samples studied are displayed in the stacked plot in Figure 8. The patterns identified included re-flections of the face-centered cubic (FCC) α-Al phase, represented by Al(111), (200), (220), (311), (222), and Al-Li phases, based on its main line (022) with relatively lower detection peak. The XRD patterns of the samples after the addition of Ti are shown in the inset of Figure 8 and display broadened and shifted Al(111) peaks. The broadening and shift are attributed to lattice defects such as internal strains, dislocations, small crystal sites, and additional grain boundaries. According to the XRD results, no new phases were discovered as a result of the titanium modification. This is likely due to the low content of titanium as an alloying element. Additionally, it could be a result of a very low quantity of newly formed phases that is below the detection limit of the XRD method.

### 3.3. Corrosion Resistance Tests

Figure 9 shows the open circuit diagram of AlLi6% in its initial state and with various concentrations of Ti after an exposure time of 1 h. The Eocp values for the samples in their initial state remained almost unchanged over time. Additionally, some fluctuations in the Eocp changes were observed, which could indicate instability of the passive layer on the surface of Sample B under the test conditions. Similarly, fluctuations in the Eocp progress were also observed for samples T3 and T4. However, the Eocp values for both samples gradually decreased over the test time, without any visible stabilization. A decrease in Eocp suggests the possibility of dissolution of the material surface. Only in the case of sample T2, the Eocp values initially shifted towards more positive values over time, indicating an increase in the compactness of the passive layer or corrosion products on the sample over time. For sample T1, the Eocp values initially decrease rapidly up to 250 s and then gradually increased over time. The highest open circuit potential value was recorded for the samples in the initial state and the average was close to −770 mV vs. Ag/AgCl. For all samples with various concentrations of Ti, the Eocp values were lower. However, the lowest Eocp value was recorded for sample T1 (0.05 wt% Ti). It was found that as the concentration of titanium in the Al-Li alloy increased, the Eocp values also increased. The Eocp value for sample T4 (1 wt% Ti) was found to be similar to that of the samples in their initial state.

The results of the pitting corrosion test in the form of anodic polarization curves are shown in Figure 10, and the values that describe the corrosion resistance of the tested samples are listed in Table 2. The recorded corrosion potential values (Ecorr) match the behavior of the open circuit potential (Eocp). The highest values of Ecorr were found in the samples in their initial state, and the lowest in sample T1—as the concentration of titanium increases, the Ecorr values also increase. However, the results of the experiment showed that sample T3 had an Ecorr value that was close to −1100 mV vs. Ag/AgCl, which was lower than the samples with lower titanium concentrations. The data also revealed that the cathodic polarization curve shifted towards higher current densities with the addition of titanium to the Al-Li alloy, indicating that the surface of the alloy with titanium had a higher rate of hydrogen evolution compared to the Al-Li alloy without titanium. As stated in the research of El-Sayed et al. [47], the increased reaction of hydrogen evolution on the surface of the alloy with Ti can be attributed to the lower hydrogen over-potential on the Al_3_Ti particles compared to the Al surface. Only in the case of sample T4, there was a slight increase in the cathodic slope compared to the sample in its original state.

The results of the pitting corrosion tests revealed that all of the tested samples exhibited similar corrosion mechanisms. It was observed that all samples had a hysteresis loop and in-effect breakdown potential, indicating the initiation and development of pitting corrosion. The only exception was sample T3, which showed signs of repassivation, indicating a reconstruction of the passive layer. Additionally, the values of corrosion current density (icorr) were found to be higher for the AlLi-Ti samples compared to the initial sample, indicating that the rate of corrosion increased as the amount of Ti in the specimen increased. The corrosion current density, or icorr, was found to be the lowest in the sample in their original state, before the addition of Ti, with an average value of around 0.31 µA/cm^2^. However, the icorr values for the AlLi-Ti alloys (excluding sample T2) were significantly higher, at over 15 times that of the initial state. An exception was sample T2, which had an icorr value of around 32 µA/cm^2^, which was 100 times higher than the initial state. This significant increase in icorr for sample T2 is thought to be due to the presence of casting defects such as pores on the surface.

### 3.4. Hardness

At room temperature, the Vickers microhardness of the prepared samples was investigated, to assess the mechanical performance of the alloy after the addition of the Ti alloying element. Figure 11 shows the results of the microhardness tests carried out, taking into account all surface conditions after metallographic polishing, in the form of hardness maps. The least microhardness value was recorded for the base alloy before the addition of Ti (Figure 11a). This sample had an average microhardness of 49.4 HV. In Figure 11b,c, a significant increase in the microhardness was observed for samples T1 and T2 after alloying with 0.05 wt% Ti and 0.1 wt% Ti, respectively. Sample T1 recorded an average microhardness of 91.8 HV, a minimum value of 85.1 HV, a maximum of 99.2 HV, and a standard deviation (SD) of 2.8. Sample T2 similarly recorded an average microhardness of 91.8 HV, having a minimum value of 79.0 HV, a maximum of 101.9 HV, and a standard deviation of 4.0. Further increase in the Ti content to 0.5 wt% in Sample T3 recorded an average microhardness of 86.2 HV, a decline of about 6% from T2. For sample T4, containing 1 wt% Ti, the average microhardness was 89.8 HV, with a standard deviation of 4.0. The outcomes of microhardness measurements on ten specifically chosen points of each sample’s surface are presented in Figure 12. The goal was to determine the surface microhardness while ignoring any surface defects. A similar pattern in hardness was found for all samples as was earlier observed in the results presented in Figure 11. The base material had the lowest average hardness (50.5 HV), while the samples containing Ti experienced a rise in average hardness by a minimum of 44%. Sample T2 had the highest mean hardness of 97.2 HV and a standard deviation of 2.0 in this case. Based on these hardness tests, it can be concluded that any irregularities on the surface of the samples resulting from the casting process did not significantly affect their overall hardness.

The effect of adding Ti to the alloy on its microhardness was found to be directly related to the microstructural changes in the alloy, such as grain size and refinement [48]. The samples T1 and T2 had the smallest grain sizes after the addition of Ti (as shown in Figure 3), resulting in the highest microhardness values. On the other hand, as the content of Ti increased in samples T3 and T4, their average grain size increased (as shown in Figure 3), leading to a decrease in the microhardness values. Materials with smaller grain sizes have more grain boundaries, which effectively impede the motion of dislocations during indentation more efficiently than materials with larger grain sizes. The observed increase of about 46% in the microhardness of the alloy after addition of Ti in this work, is supported by similar research conducted by Zhuang et al. [49], who studied the effect of titanium alloying on the microstructure and properties of high manganese steel. It was reported from that research that the addition of Ti facilitated grain refinement of the alloy and increased the hardness by 87%. Similar studies [50,51,52,53,54,55] have also reported observed increase in the microhardness of aluminum alloys upon alloying with Ti. They argue that the observed increase in the alloy’s hardness is influenced by the refined microstructure obtained by adding small amounts of Ti element, and with an increase in the content of Ti element, there is nucleation of peritectic intermetallic Al_3_Ti, which further increases the hardness of the alloy.

## 4. Conclusions

In this study, Al-Li alloys were created with added titanium contents to investigate the impact on the microstructure, corrosion resistance, and hardness. The base alloy consisted of Al and 6% Li, and different amounts of titanium were added to make the samples at 0.05 wt%, 0.1 wt%, 0.5 wt%, and 1.0 wt%. Based on the research, the following conclusions were drawn.

The microstructure of the alloy was modified after adding Ti, resulting in a decrease in average grain size to about 60% with the optimal refinement happening at 0.05 wt% Ti content. However, further increases in Ti content resulted in an increase in average grain size.SEM observations, coupled with EDS analyses revealed an irregular net-shaped interdendritic microstructure, with an elemental microsegregation at the grain boundaries, mostly made up of Li.Casting defects were observed after sample preparation, attributed to the high content of Li in the alloy, which absorbed air during casting, resulting in material defects such as shrinkage holes. The corrosion resistance test results were correspondingly low for the samples showing casting defects. The sample T2, with more casting defects, consequently recorded the least corrosion resistance.The microhardness of the alloy increased from an average of 49.4 ± 7.7 HV to the highest average of 91.8 ± 2.8 HV, after the addition of the Ti alloying element. This is attributed to the structural grain refinement of the alloy and the formation of Al_3_Ti intermetallic, upon the addition of Ti to the base Al-Li alloy.

## Figures and Tables

**Figure 1 materials-16-02671-f001:**
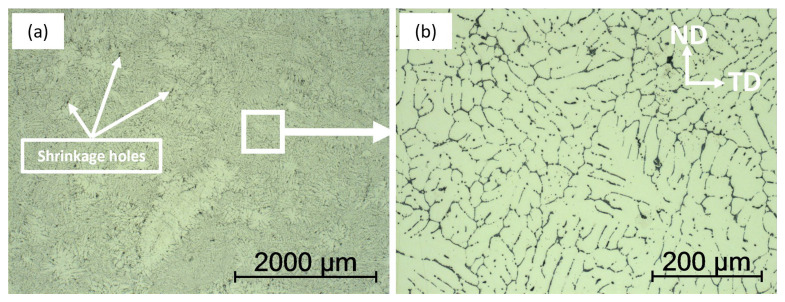
Microstructure of as-cast base Al-6Li before addition of Ti alloying elements (**a**) micrograph showing the structure of the sample and the shrinkage holes developed after casting (**b**) micrograph showing a dendritic network of non-equilibrium eutectic phases distributed in the grain boundaries (ND: normal direction, TD; transverse direction).

**Figure 2 materials-16-02671-f002:**
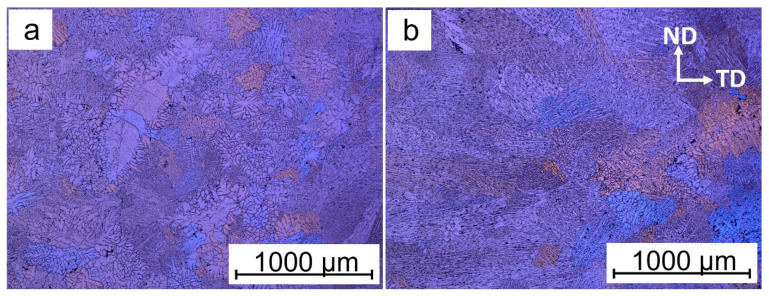
Light micrographs of investigated samples showing the microstructural grains and grain boundaries (**a**) Sample B (**b**) Sample T1 (**c**) Sample T2 (**d**) Sample T3 (**e**) Sample T4 (ND: normal direction, TD; transverse direction).

**Figure 3 materials-16-02671-f003:**
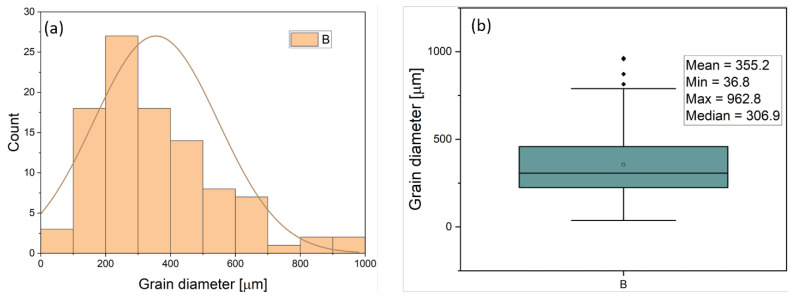
Grain size distribution plots: (**left**) grain diameter distribution curves (**right**) box and whisker charts with statistical derivations (**a**,**b**) Sample B, (**c**,**d**) Sample T1, (**e**,**f**) Sample T2, (**g**,**h**) Sample T3, (**i**,**j**) Sample T4.

**Figure 4 materials-16-02671-f004:**
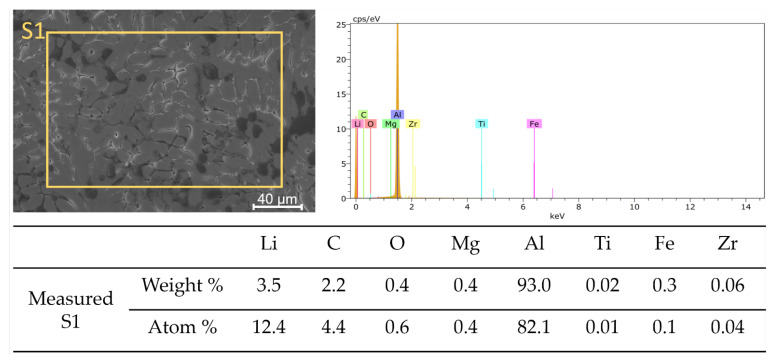
Energy-dispersive X-ray spectrometry (EDS) chemical composition analysis of sample T1 after addition of Ti alloying element.

**Figure 5 materials-16-02671-f005:**
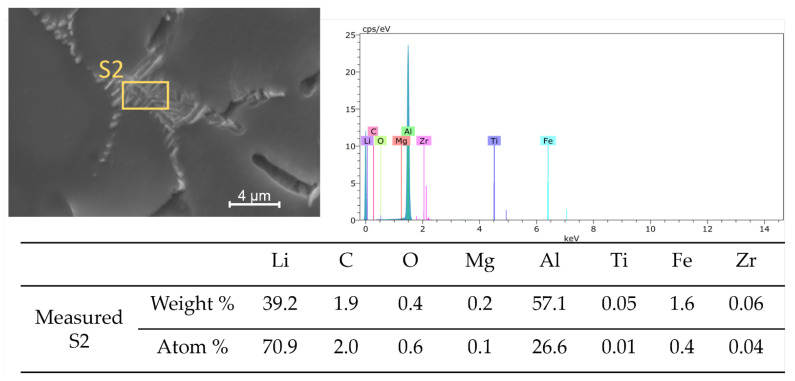
EDS chemical composition analysis of the elemental microsegregation at the grain boundary.

**Figure 6 materials-16-02671-f006:**
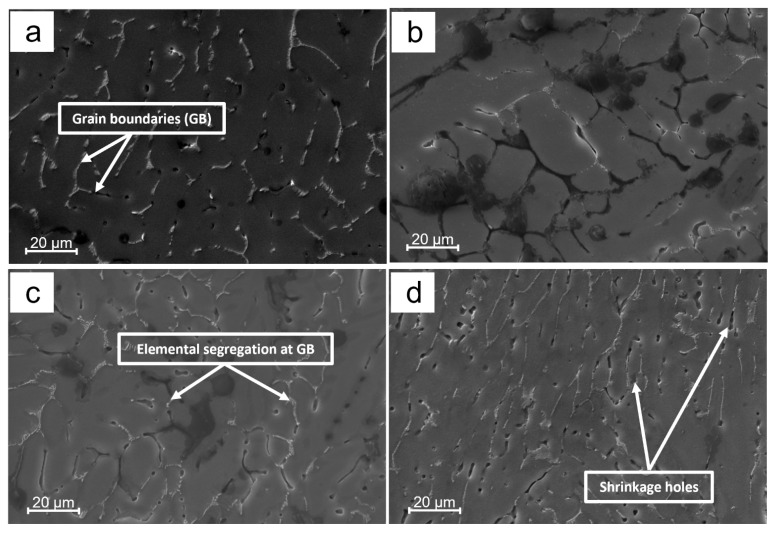
SEM images of investigated samples (**a**) Sample B (**b**) Sample T1 (**c**) Sample T2 (**d**) Sample T3 (**e**) Sample T4.

**Figure 7 materials-16-02671-f007:**
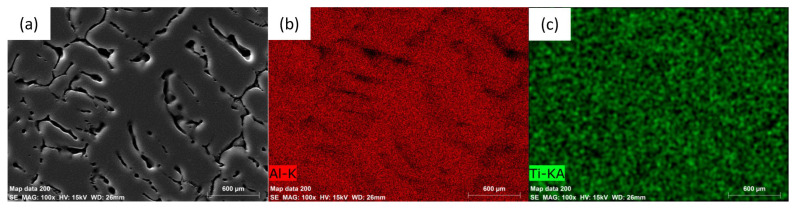
Energy-dispersive X-ray spectrometry (EDS) maps of sample T2 showing the distribution of Al and the Ti alloying element in the sample (**a**) area under observation, (**b**) distribution of Al over the observed area, (**c**) distribution of Ti over the observed area.

**Figure 8 materials-16-02671-f008:**
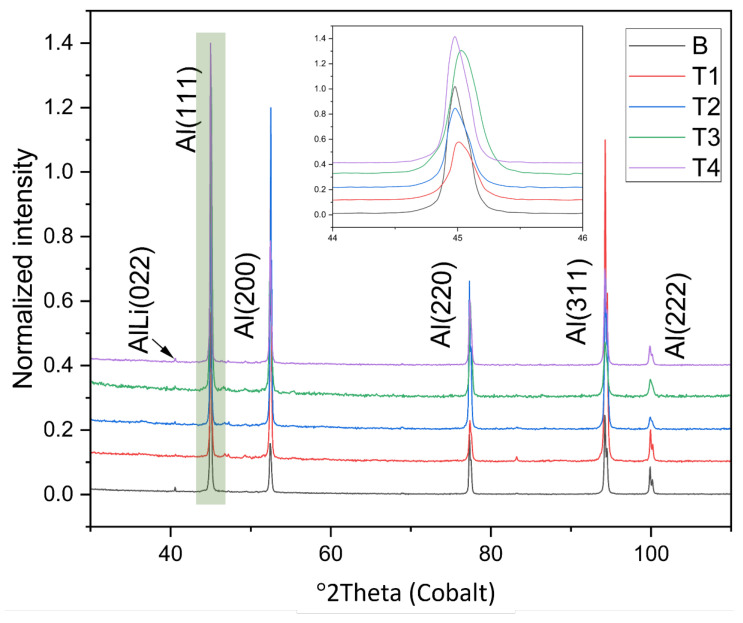
XRD patterns of investigated Al-Li alloys.

**Figure 9 materials-16-02671-f009:**
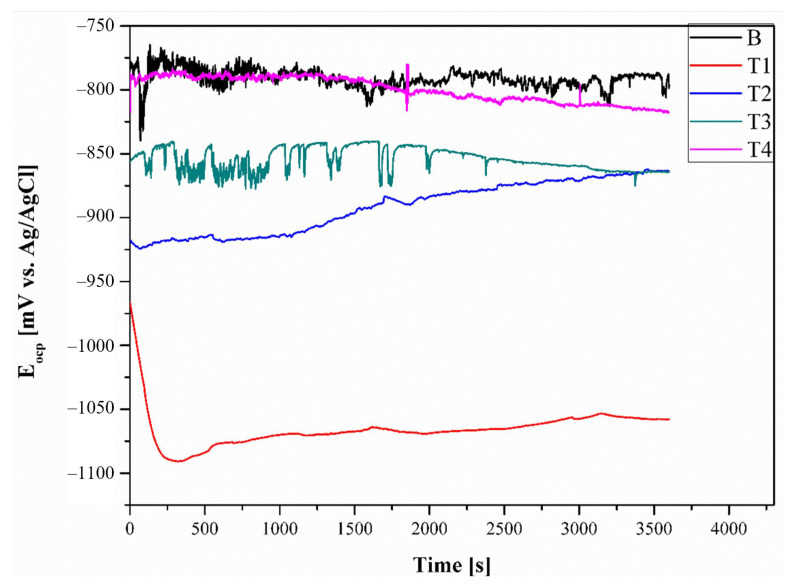
Open circuit potential changes in t = 1 h.

**Figure 10 materials-16-02671-f010:**
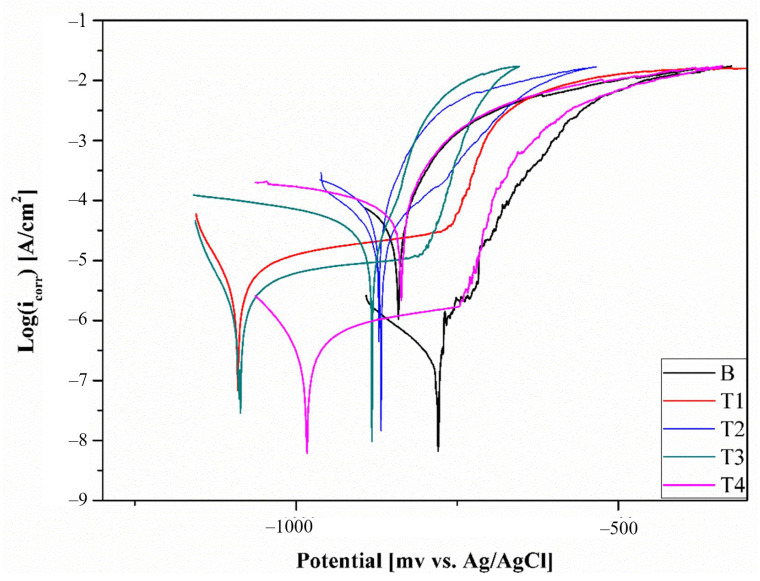
Potentiodynamic polarization curves of tested samples.

**Figure 11 materials-16-02671-f011:**
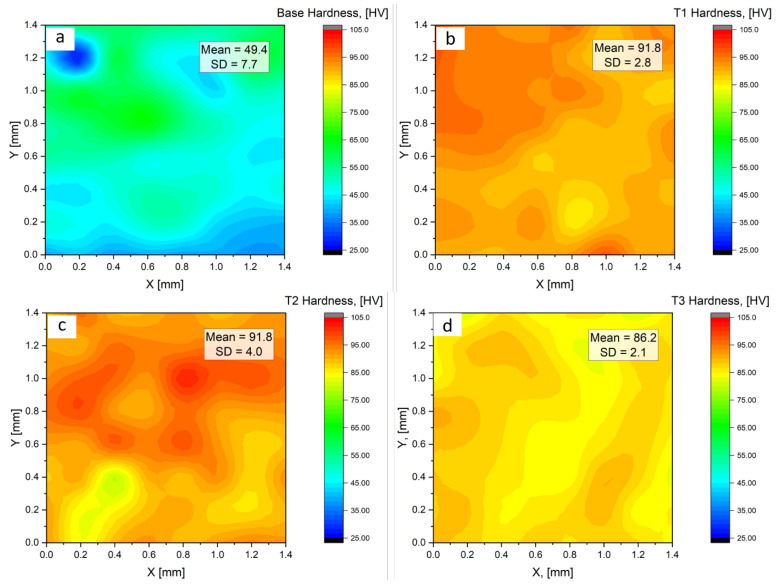
Vickers microhardness maps for studied samples, considering all surface conditions over an area of 1.4 mm × 1.4 mm (**a**) Sample B (**b**) Sample T1 (**c**) Sample T2 (**d**) Sample T3 (**e**) Sample T4.

**Figure 12 materials-16-02671-f012:**
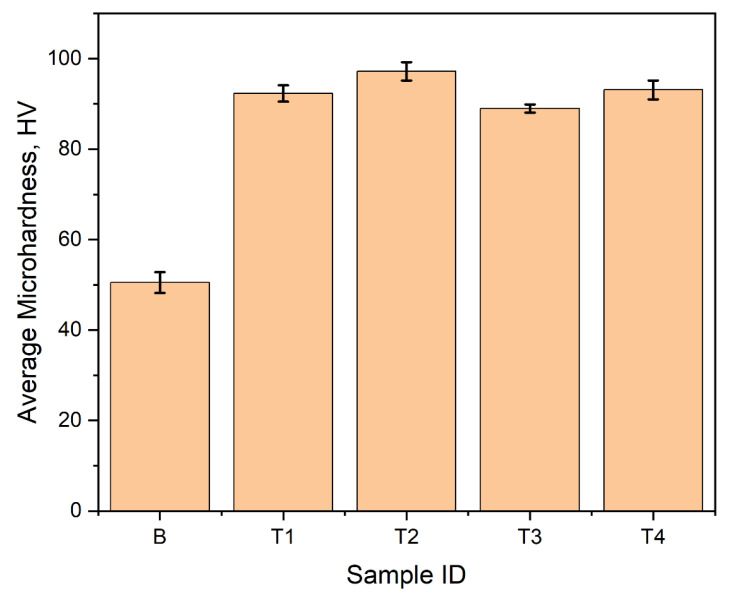
Vickers microhardness of investigated materials samples.

**Table 1 materials-16-02671-t001:** Designation of prepared samples for investigations and their respective constituent elements for fabrication.

Sample ID	Alloy Constituents
B	Al+6%Li
T1	Al+6%Li + 0.05 wt% Ti
T2	Al+6%Li + 0.1 wt% Ti
T3	Al+6%Li + 0.5 wt% Ti
T4	Al+6%Li + 1.0 wt% Ti

**Table 2 materials-16-02671-t002:** Results of corrosion test.

Sample ID	E_corr_ [mV vs. Ag/AgCl]	E_b_ [mV vs. Ag/AgCl]	E_np_ [mV vs. Ag/AgCl]	i_corr_ [µA/cm^2^]
B	−798	−616	-	0.36
T1	−1105	-	-	4.56
T2	−863	−705	-	32.44
T3	−1100	−741	−882	4.97
T4	−811	−790	-	5.22

## Data Availability

Not applicable.

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
