# Peer review of "Impact of Titanium Addition on Microstructure, Corrosion Resistance, and Hardness of As-Cast Al+6%Li Alloy"

_materials, 2023, doi:10.3390/ma16072671_

Round 1

Reviewer 1 Report

I have the following comments on the submitted article:

1. delete the entire paragraph on lines 48 - 53

2. I recommend shortening the introduction by at least 1/3

3. I appreciate the statistical treatment of grain size from the point of view of metallography in Fig. 2. As well as in Fig. 3 and the overall summary in Tab. 2. I recommend the authors to keep only Fig. 2 and either Fig. 3 or Tab. 2. Because the graphs in Fig. 2 and Tab. 2 describe more or less the same values.

In the case of keeping fig. 3, I recommend expanding the text description of the X and Y axes

4. Tab. 3. remove, its values can be easily read in Fig. 9 (Unnecessary repetition of results)

5. Repeating the results in the form of pictures and graphs. I recommend entering the values from tab. 5 to the created microhardness maps depending on the sample.

The values of microhardness on samples T1, T2 and T4 overlap within the standard deviation, so it is not possible to conclude that the microhardness of the examined samples increases with the gradual reduction of grain size.

Author Response

Reviewer #1

  • Comment: delete the entire paragraph on lines 48 – 53
  • Response: The said paragraph has now been deleted.

  • Comment: I recommend shortening the introduction by at least 1/3
  • Response: The introduction has been shortened. The word count has been reduced from 1296 to 948.

  • Comment: I appreciate the statistical treatment of grain size from the point of view of metallography in Fig. 2. As well as in Fig. 3 and the overall summary in Tab. 2. I recommend the authors to keep only Fig. 2 and either Fig. 3 or Tab. 2. Because the graphs in Fig. 2 and Tab. 2 describe more or less the same values. In the case of keeping fig. 3, I recommend expanding the text description of the X and Y axes.
  • Response: Table 2 has been removed. The authors decided to keep figure 3, so the text descriptions on the X and Y axes have been expanded.

  • Comment: 3. remove, its values can be easily read in Fig. 9 (Unnecessary repetition of results)
  • Response: Table 3 has been removed.

  • Comment: Repeating the results in the form of pictures and graphs. I recommend entering the values from tab. 5 to the created microhardness maps depending on the sample.

The values of microhardness on samples T1, T2 and T4 overlap within the standard deviation, so it is not possible to conclude that the microhardness of the examined samples increases with the gradual reduction of grain size.

  • Response: The results previously presented in table 5 have been entered in the hardness maps. The authors acknowledge the overlap in the microhardness values of the samples containing Titanium. The hardness experiment has been repeated, and this time, measurements were made on surface points devoid of imperfections. These results are now presented in figure 12. The discussions in section 3.3 have correspondingly been updated to reflect this update for clarity.

Reviewer 2 Report

1) The first question that arises when reading this work is: why was the concentration of 6% Li chosen when the introduction refers to a much lower concentration? It was justly noted by the authors at the very beginning that the addition of Li increases the inhomogeneity of the structure. Why did the authors choose 6%?

2) How was the melting temperature of the investigated alloys chosen? Was it the same for all alloys? If so, then the alloys were obtained with different overheating above the liquidus, which significantly affects the properties.

3) “The carbon 222 compounds formed by Titanium act as cores for heterogenous nucleation, promoting the 223 nucleation of molten Al3Li. This thereby influences the further growth of inclusions and 224 grains, significantly reducing or increasing the grain size, depending on the Ti content. 225 [38] 226» First, there should be a clear dependence of the grain size on the particle concentration, therefore, with an increase in the titanium concentration, the grain size should decrease. Secondly, reference 38 refers to steels.

4) Where did C, Mg, Zr come from in Figures 4 and 5, if you received from pure metals?

5) “• SEM observations, coupled with EDS analyzes revealed an irregular netshaped interdendritic microstructure, with an elemental microsegregation at 394 the grain boundaries” Was the analysis performed on samples without prior homogenization? In general, how were these alloys thermally processed after casting?

6) "The microhardness of the alloy increased from an average of 49.4 ± 7.7 HV 403 to a highest average of 91.8 ± 2.8 HV, after the addition of Ti alloying ele404ment" what caused this increase?

What is the final conclusion of this work? The information that it is not desirable to add a lot of Li to aluminum alloys was known at the very beginning of the work, you only confirmed these data

Author Response

Reviewer #2

  • Comment: The first question that arises when reading this work is: why was the concentration of 6% Li chosen when the introduction refers to a much lower concentration? It was justly noted by the authors at the very beginning that the addition of Li increases the inhomogeneity of the structure. Why did the authors choose 6%?
  • Response: The continuous interest in aluminum-lithium alloys, which are characterized by a low density and high modulus of elasticity, is attributed to the advantageous properties of lithium. With each percent of lithium added, the density of aluminum decreases by 3%, and the elastic modulus increases by 6%. Additionally, these alloys exhibit significant hardening effects after thermal treatment and artificial aging. The aluminum-lithium phase diagram (shown in the figure) summarizes the possible and conducted research on this system, revealing that the eutectic line towards aluminum passes at 7,5 % and 602°C. At the eutectic temperature, the solubility of lithium in aluminum is 4,2 % and reaches approximately 18 %. It is worth noting that the concentration of 6% lithium was selected based on the Al-Li diagram in order to obtain low-density alloys, and the addition of Ti serves as the microstructure refining factor.

Fig. - Diagram of the Al-Li system.

  • Comment: How was the melting temperature of the investigated alloys chosen? Was it the same for all alloys? If so, then the alloys were obtained with different overheating above the liquidus, which significantly affects the properties.
  • Response: Thank you for this remarks, similar, the authors were guided by the Al-Ti phase system in selecting the melting temperature. The Authors assumed that with low concentration of Ti - max 1 %, the chosen melting temperature allows for close condition of cast preparation. The authors have to confirm that all samples were subjected to the same conditions.

  • Comment: The carbon compounds formed by Titanium act as cores for heterogenous nucleation, promoting the nucleation of molten Al3Li. This thereby influences the further growth of inclusions and grains, significantly reducing or increasing the grain size, depending on the Ti content.  [38]» First, there should be a clear dependence of the grain size on the particle concentration, therefore, with an increase in the titanium concentration, the grain size should decrease. Secondly, reference 38 refers to steels.
  • Response: The highlighted discussion has been revised for clarity. The cited reference has been replaced with another which is more in line with the current work.

  • Comment: Where did C, Mg, Zr come from in Figures 4 and 5, if you received from pure metals?
  • Response: The identified trace elements Mg, Zr and C are attributed to the inclusions resulting from the casting process. According to the EDS analysis, their quantities are quite low such that they do not significantly affect the results discussed in this work. Carbon concentration additionally can be attributed to the contamination of the EDS system.

  • Comment: “SEM observations, coupled with EDS analyzes revealed an irregular netshaped interdendritic microstructure, with an elemental microsegregation at the grain boundaries” Was the analysis performed on samples without prior homogenization? In general, how were these alloys thermally processed after casting?
  • Response: In this version of the paper, all investigations were carried out on the as-cast samples as reported in other publications [1–4], on the performance of aluminum alloys upon addition of different alloying element. The title of the paper has been updated to reflect this.
  1. Wu, L.; Li, X.; Wang, H. The Effect of Major Constituents on Microstructure and Mechanical Properties of Cast Al-Li-Cu-Zr Alloy. Mater. Charact. 2021, 171, 110800, doi:10.1016/j.matchar.2020.110800.
  2. Zhang, J.; Wu, G.; Zhang, L.; Zhang, X.; Shi, C.; Sun, J. Effect of Zn on Precipitation Evolution and Mechanical Properties of a High Strength Cast Al-Li-Cu Alloy. Mater. Charact. 2020, 160, 110089, doi:10.1016/j.matchar.2019.110089.
  3. Chen, K.; Li, Z.; Cao, Y.; Wu, X.; Chen, X.; Wang, Z.; Tian, Y.; Pan, S. Effect of Composition on Microstructure and Properties of As-Cast 2196 Al–Cu–Li Alloy. Mater. Sci. Technol. 2022, 38, 1168–1184.
  4. Rangel-Ortiz, T.; Chávez-Alcalá, F.; Curiel-Reyna, E.; Real, A. del; Baños, L.; López-Hirata, V.M.; Rodríguez, M. Structural and Mechanical Characterization of As-Cast Al–Li–Hf Alloy. Mater. Manuf. Process. 2007, 22, 247–250.

  • Comment: "The microhardness of the alloy increased from an average of 49.4 ± 7.7 HV 403 to a highest average of 91.8 ± 2.8 HV, after the addition of Ti alloying element" what caused this increase?
  • Response: This part of the conclusion has been revised to include the cause of the increase in the hardness after addition of Ti alloying element.

  • Comment: What is the final conclusion of this work? The information that it is not desirable to add a lot of Li to aluminum alloys was known at the very beginning of the work, you only confirmed these data.
  • Response: This work focused on improving the performance of low-density Al-Li alloys by addition of Titanium. The novelty lied in the use of combination of 6% Li and Ti microalloying, which has not been extensively studied in literature. However, whiles the selection of this high content of Li was aimed at reducing the density of the alloy with Li, the inhomogeneities introduced by the use Li could impact the performance of the alloy. Titanium, which is widely regarded as an efficient inoculant for aluminum alloys grain refinement was used to alloy this high Li content Al-Li alloy with the aim of enhancing its mechanical properties and corrosion properties. This research, therefore, presented the results obtained in this regard.

Round 2

Reviewer 1 Report

Thanks to the authors for editing the article according to my instructions.

I recommend the article corrected in this way for publication.

Author Response

We are pleased to hear that our revised manuscript has been accepted following the corrections that were made based on your insightful feedback. We appreciate the time and effort you have put into reviewing our work and providing valuable suggestions for improvement.

Reviewer 2 Report

"Response: Thank you for this remarks, similar, the authors were guided by the Al-Ti phase system in selecting the melting temperature. The Authors assumed that with low concentration of Ti - max 1%, the chosen melting temperature allows for close condition of cast preparation. The authors have to confirm that all samples were subjected to the same conditions."

The choice of alloy casting temperature remains unclear. It is likely that the alloys were not obtained under the same conditions.

it is necessary to conduct additional experiments and provide additional results such as differential scanning calorimetry (DSC)

Author Response

The authors sincerely appreciate the reviewer's comments and suggestions, which are highly valuable in improving the quality of the manuscript.

In order to clarify the choice of casting process condition the authors have updated the experimental design discussed in Section 2 to include more insights into the casting process. We confirm that our goal was to fabricate Al-Li alloys with very low density while improving their performance by using low concentrations of Ti (max 1%) in the alloying process.

While we acknowledge that additional experiments such as differential scanning calorimetry (DSC) would provide more information to support the discussed results, we were limited by resources to conduct this experiment in this version of the manuscript. However, we are confident that the primary objective of the paper, which is to investigate the corrosion and hardness properties of the as-cast alloys, has been achieved.

We thank the reviewer for suggesting further experiments, and we will certainly take this into account in future works to provide more insights into these alloys.

Once again, we would like to express our gratitude to the reviewer for their valuable feedback and insights, which have helped us to improve the manuscript.